# CSD: Content-aware Speculative Decoding for Efficient Image Generation

**Mingcheng Wang**[1] **Junbo Qiao**[1] **Yunchen Li**[1] **Lingfu Jiang**[1] **Wei Li**[2] **Jie Hu**[2] **Jiao Xie**[1] **Zhou Yu**[1 3]
**Xinghao Chen**[2] **Guixu Zhang**[1] **Shaohui Lin**[1 3]

## Abstract

Speculative decoding (SD) has emerged as a key solution to accelerate the inference of autoregressive models. However, in the field of image generation, it faces the challenge of low acceptance rates, and directly relaxing its criteria leads to degradation in image quality. In this paper, we propose a novel content-aware speculative decoding algorithm, termed CSD, which integrates an entropy-based probability relaxation mechanism with an optimal resampling strategy to enhance the inference efficiency for autoregressive image generation. By leveraging the informational uncertainty inherent in different regions of an image, CSD dynamically adjusts the acceptance probability of candidate tokens, increasing the acceptance rate in low-detail areas to accelerate generation. Moreover, a distribution alignment filter is introduced to ensure the output distribution to be aligned with the target model, which significantly improves the generative quality. Experiments conducted on Lumina-mGPT and Janus-Pro demonstrate that the superiority of the proposed CSD. Our source code is available at https://github.com/aderfebr/CSD.

## 1. Introduction

Autoregressive (AR) models have demonstrated a strong capacity for modeling complex data distributions through their token-by-token generation mechanism, as evidenced by their success in large language models (LLMs) (Achiam et al., 2023; Liu et al., 2024a; Yang et al., 2025). Inspired by this progress, recent studies have introduced the AR modeling paradigm into the domain of image generation and achieved remarkable results (Team, 2024; Liu et al.,

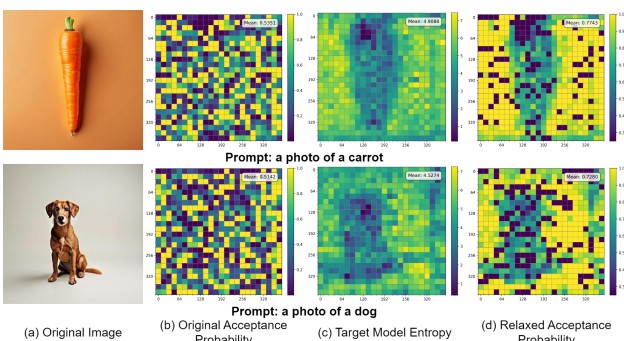

*Figure 1.* Motivation of CSD. (a) Original image. (b) The original acceptance probability of SJD shows no rules to generate image, which is typically content-agnostic. (c) The entropy of target model shows the image intrinsic pattern, which can be used to reduce the computation of smooth regions by adding to the acceptance probability. (d) The relaxed acceptance probability has a high probability to reduce the computation on the smooth regions for fast decoding.

2024b; Chen et al., 2025). These advances position AR image models as competitive alternatives to diffusion-based methods. Moreover, AR-based image generation models exhibit several unique advantages, such as natural support for multimodal generation (Team, 2024).

Despite these benefits, autoregressive image generation suffers from a fundamental limitation via strictly sequential decoding. Generating a high-resolution image requires the model to produce thousands of tokens one by one, which significantly increases the computation cost for image generation. For example, on a single NVIDIA A6000 GPU, Janus-Pro 7B (Chen et al., 2025) requires an average inference time of 22.95s to generate a single image of $384 \times 384$ resolution, while Lumina-mGPT 7B (Liu et al., 2024b) requires approximately 121.48s to produce an image at a higher resolution of 768×768. This efficiency problem of AR models has become a major obstacle to the practical deployment, especially in resource-limited applications.

To overcome the decoding inefficiency of autoregressive image generation, non-autoregressive (NAR) approaches ( e.g., MaskGIT (Chang et al., 2022) and AutoNAT (Ni et al., 2024a)) aim to generate images via parallel decoding and mask prediction. However, their parallel generation capabil-

[1]East China Normal University [2]Huawei Foundation Model Dept [3]Key Laboratory of Advanced Theory and Application in Statistics and Data Science-MOE. Correspondence to: Shaohui Lin <shlin@cs.ecnu.edu.cn>, Zhou Yu <zyu@stat.ecnu.edu.cn>.

*Proceedings of the $43^{rd}$ International Conference on Machine Learning*, Seoul, South Korea. PMLR 306, 2026. Copyright 2026 by the author(s).

ity stems from their inherent design paradigm and cannot be directly transferred to autoregressive generation without fundamentally altering the model architecture. Recently, speculative decoding (SD) based multi-token prediction has garnered significant attention due to its inherent flexibility. SD uses a small draft model to predict multiple candidate tokens, which are verified in parallel by the large target model. Obviously, SD requires less time to run a large target model, compared to the traditional one-by-one token prediction of AR. For example, EAGLE (Li et al., 2024) trains a lightweight auxiliary model to achieve multi-token prediction, while SJD (Teng et al., 2024) utilizes Jacobi decoding to generate multiple tokens without additional models. Although existing approaches accelerate image generation, these speculative decoding strategies are content-agnostic. The background in the top figure in Figure 1 (a) shows the relatively uniform pattern, occupied with a significant area, yet the acceptance rate using SJD in Figure 1(b) does not differ substantially from that in the bottom figure. Moreover, there is no pattern to the token acceptance, making it impossible to effectively explain the token selection mechanism of SJD.

Actually, natural images often exhibit significant heterogeneity, comprising both smooth regions and textured areas. Smooth regions tend to have higher redundancy, making them easier to predict and particularly well-suited for probabilistic guessing in speculative decoding. In contrast, fine-grained textures are far less predictable and considerably more challenging to generate accurately. We found entropy can serve as a measure of confidence in LLM reasoning (Kang et al., 2025). We assume that this entropy-based confidence is often closely correlated with texture-related information for autoregressive image generation. As such, we perform a fine-grained analysis of content-dependent factors in autoregressive image generation. As illustrated in Figure 1(c), the average entropy of the target model in the top image is higher than that in the bottom image. Moreover, regions with higher entropy strongly correlate with background areas, whereas regions with lower entropy correspond to fine-grained textures. This finding raises a key question: *Can entropy be leveraged to design a content-aware speculative decoding for more efficient and rational image generation?*

To answer the above question, we propose a **Content-aware Speculative Decoding** algorithm, termed *CSD*, which leverages the entropy of the target model as a guiding signal to relax the acceptance probability, thereby enabling better acceleration in smooth regions. Under the content-aware probabilistic relaxation, the resampling distribution used in the original speculative decoding remains optimal. Furthermore, we introduce a distribution alignment filter based on total variation (TV) distance to determine when probabilistic relaxation should be applied, thereby further mitigating poten-

tial degradation in generation quality. Figure 1(d) presents the acceptance probabilities after processing, which have been mapped for better visualization. It can be observed that the post-processed acceptance probabilities are highly correlated with target model entropy, achieving content-aware speculative decoding. Extensive experiments demonstrate that the proposed CSD enables content-aware adaptive acceleration to outperform existing SD approaches in terms of both efficiency and generation performance.

Our main contributions are summarized as below:

- We propose a novel content-aware speculative decoding algorithm, which is implemented by leveraging the entropy of the target model as a confidence criterion for the probabilistic relaxation in SD. Moreover, we prove that the corresponding resampling distribution can be regarded as an optimal resampling distribution.

- We design a distribution alignment filter to reduce the discrepancy between the generated sequence and the target one, improving the image generation quality.

- Extensive experiments show the superiority of the proposed CSD. For example, we accelerate the inference of Janus-Pro (Chen et al., 2025) by 4.33× with only a negligible CLIP score drop on the MS-COCO benchmark (Lin et al., 2014), outperforming the SOTA GSD (So et al., 2025).

## 2. Related Work

### 2.1. Autoregressive Image Generation

Early pixel-level autoregressive image models, such as PixelRNN, PixelCNN (Van Den Oord et al., 2016), and PixelCNN++ (Salimans et al., 2017), generate images by raster scanning and sequentially predicting each pixel. DALL·E (Ramesh et al., 2021) introduces a token-based paradigm that encodes images into discrete visual tokens via VQ-VAE (Van Den Oord et al., 2017) and models them using large-scale autoregressive transformers. LlamaGen (Sun et al., 2024) further adapts the next-token prediction framework from large language models to image generation. Recent efforts have explored unified multimodal autoregressive modeling. Chameleon (Team, 2024) enables mixed processing and generation of images and text within a single model. Lumina-mGPT (Liu et al., 2024b) builds upon this foundation with multimodal generative pre-training and progressive supervised fine-tuning. Janus-Pro (Chen et al., 2025) further improves generation quality through optimized training strategies and large-scale data, achieving state-of-the-art performance among autoregressive image models. However, these methods are constrained by the inherent of autoregressive image decoding. This property makes them difficult to deploy in latency-sensitive scenarios.

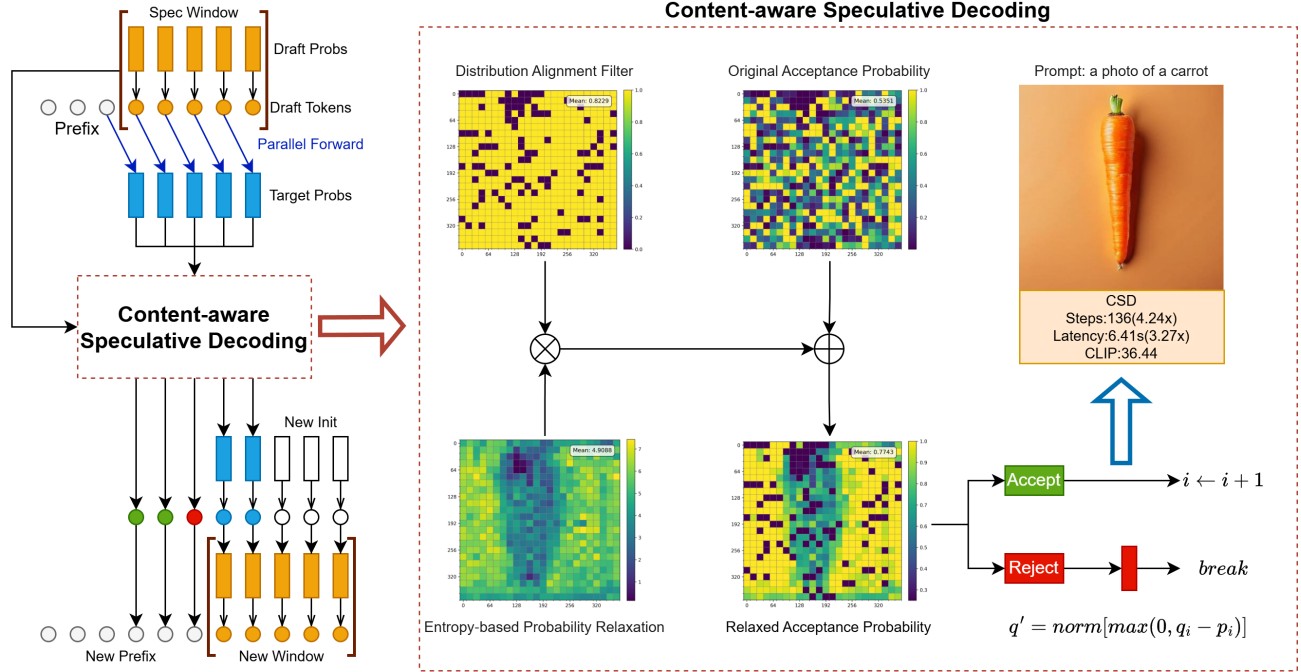

*Figure 2.* Overview of the proposed CSD framework. It consists of two key components: (1) an entropy-based probability relaxation module that dynamically increases acceptance probability in low-detail regions, and (2) a distribution alignment filter based on TV distance, which preserves output quality by excluding tokens where the distribution discrepancy is large.

## 2.2. Speculative Decoding for Image Generation

Speculative Decoding (SD) (Leviathan et al., 2023) is an effective technique to accelerate autoregressive decoding. It employs a lightweight draft model to produce candidate tokens, which are then verified in parallel by a target model to preserve the distributional consistency. However, suitable off-the-shelf draft models are often unavailable for autoregressive image generation. Recent work has explored dependent draft approaches. EAGLE (Li et al., 2024) trains an auxiliary model to predict changes in the target model's hidden states. LANTERN (Jang et al., 2024) further aggregates token probabilities within clusters while constraining the total variation distance. SJD2 (Teng et al., 2025) introduces a next-clean-token prediction paradigm to serve as a draft model. While these training-based methods achieve substantial step compression with high visual quality, they incur additional overhead during both training and inference.

In contrast, training-free acceleration methods have also been investigated. Jacobi Decoding (Song et al., 2021) enables parallel updates by iteratively refining draft tokens until convergence. SJD (Teng et al., 2024) integrates Jacobi Decoding with speculative decoding to achieve lossless acceleration. ZipAR (He et al., 2024) exploits spatial locality to enable parallel decoding across image rows. GSD (So et al., 2025) relaxes acceptance criteria by grouping tokens based on probability and embedding similarity. Unlike these methods, our approach focuses on content-aware specula-

tive decoding and dynamically adapts the decoding strategy based on image content, without introducing additional training overhead.

## 2.3. Content-adaptive Generation

In the field of image generation, research efforts have been directed toward adaptive generation methods. For Non-Autoregressive Transformers, AutoNAT (Ni et al., 2024a) constructs an automated framework capable of systematically searching for the optimal strategy combinations tailored to specific NAT models and datasets. AdaNAT (Ni et al., 2024c) employs reinforcement learning to train a lightweight policy generation network, which automatically configures a tailored generation strategy for each sample. ENAT (Ni et al., 2024b) achieves a notable improvement in model performance while significantly reducing computational costs by independently encoding visible tokens and prioritizing the computation and updating of representations for a few critical tokens. In the Diffusion Transformer architecture, RelaCtrl (Cao et al., 2025) customizes the positioning, parameter scale, and modeling capacity by evaluating the ControlNet Relevance Score. DiffCR (You et al., 2025) addresses the inherent variation in importance among different image tokens by proposing an automated learning framework that dynamically allocates computational resources across layers and timesteps. However, these methods are incompatible with next-token-prediction paradigms

and often require additional training overhead.

## 3. Method

### 3.1. Preliminaries on Speculative Decoding

Speculative Decoding (SD) is a technique designed to accelerate the inference of autoregressive (AR) models by leveraging two models: an accurate target model $M_q$ and an efficient draft model $M_p$. Given a context prefix $X$, the method first uses the draft model $M_p$ to generate $\gamma$ candidate tokens autoregressively, yielding distributions $p_k(\cdot)$ from draft model and $q_k(\cdot)$ from target model for each position $k$, respectively. Token generated by draft model $\hat{x}_k$ is accepted with probability $b_k := \min(1, \frac{q_k(\hat{x}_k)}{p_k(\hat{x}_k)})$. If a token is rejected, it is replaced by a new token sampled from the resampling distribution $\mathcal{P}(x) = \text{norm}\{\max(0, q(x) - p(x))\}$. These tokens (accepted ones and the resampled one at the rejection point) are then appended to the context prefix $X$, forming the new input for the next round of speculative decoding. After this step, a new batch of candidate tokens is produced, and the process repeats. Critically, this mechanism guarantees that the final output distribution exactly matches that of the target model $M_q$, while ensuring that each run generates at least one new token.

It has been proved that the aforementioned procedure can accelerate generation with no degradation in output distribution (Leviathan et al., 2023). When the expected acceptance rate is $\alpha$, the expected number of tokens generated per iteration is:

$$\frac{1 - \alpha^{\gamma+1}}{1 - \alpha}. \tag{1}$$

Considering that generating the draft tokens incurs computational overhead, let $c$ denote the ratio of the time cost per iteration between the draft model and the target model. The actual expected improvement factor is:

$$\frac{1 - \alpha^{\gamma+1}}{(1 - \alpha)(\gamma c + 1)}. \tag{2}$$

Although SD accelerate the inference process, it cannot be adaptive decoding based on image content, such that significant redundant computation is introduced in the simple, smooth regions of the generative image.

### 3.2. Framework of the Proposed CSD

As illustrated in the left part of Figure 2, following SJD (Teng et al., 2024), we adopt Jacobi Decoding (Song et al., 2021) to generate draft tokens without requiring any auxiliary model training. Specifically, draft probabilities are derived from the target model's outputs from previous iterations, making our approach training-free and plug-and-play.

---

**Algorithm 1** Content-aware Speculative Decoding

**Input:** Context prefix $X$, Draft model $M_p$, Target model $M_q$, Window size $\gamma$, Maximum length $N$
$n \leftarrow 0$
**while** $n < N$ **do**
    **for** $k = 1 : \gamma$ **do**
        $p_k \leftarrow M_p(X, \hat{x}_1, \ldots, \hat{x}_{k-1}), \hat{x}_k \sim p_k$
    **end for**
    $q_1, \cdots, q_{\gamma+1} \leftarrow M_q(X), \ldots, M_q(X, \hat{x}_1, \ldots, \hat{x}_\gamma)$
    **for** $k = 1 : \gamma$ **do**
        **if** $\text{TV}(p_k, q_k) < \delta$ **then**
            $\text{H}(q_k) = -\sum_x q_k(x) log(q_k(x))$
            $\epsilon = \lambda \text{H}(q_k)$
        **else**
            $\epsilon = 0$
        **end if**
        $b_k \leftarrow \min(1, \frac{q_k(\hat{x}_k)}{p_k(\hat{x}_k)} + \epsilon)$
        $r \sim U[0, 1]$
        **if** $r \leq b_k(\hat{x}_k)$ **then**
            $x_k \leftarrow \hat{x}_k$
        **else**
            $q' \leftarrow \text{norm}\{\max(0, q_k - p_k)\}$
            $x_k \sim q'$
            **break**
        **end if**
    **end for**
**end while**
**return** $X$

---

The right part of the figure introduces the proposed content-aware speculative decoding (CSD), which consists of two main components: Entropy-based Probability Relaxation and Distribution Alignment Filter. We incorporate the entropy of the target distribution into the original speculative decoding probability while employing an optimal resampling distribution. This enables dynamic relaxation of the acceptance probability, effectively increasing it in regions with fewer details or lower complexity. To mitigate the potential degradation caused by this probabilistic relaxation, we introduce a distribution alignment filter based on total variation (TV) distance. By excluding positions where the discrepancy between the target distribution $q$ and the draft distribution $p$ is excessively large, it effectively preserves the generation quality.

The detailed procedure is outlined in Algorithm 1. At the beginning of the iteration, the draft model is first employed to generate multiple draft probabilities, from which draft tokens are sampled. Specifically, the draft probabilities are derived from the target probabilities produced in the previous iteration, supplemented by randomly initialized probabilities. Subsequently, the target model is invoked to

obtain the target probabilities at the corresponding positions. Then, the verification stage proceeds, where the TV distance between the target probabilities and the draft probabilities is computed to measure the alignment at each position. If the distance is below a predefined threshold $\delta$, $\epsilon$ is set to $\lambda H(q_k)$ and 0 otherwise, where $H(q_k)$ and $\lambda$ denote the entropy of the target distribution at position $k$ and relaxation coefficient, respectively. Thus, we obtain the acceptance probability $\min(1, \frac{q_k(\hat{x}_k)}{p_k(\hat{x}_k)} + \epsilon)$ for this position. If the token is accepted, the verification proceeds to the next position; otherwise, the process enters the rejection phase. In the rejection phase, a token is resampled from the normalized distribution $\text{norm}\{\max(0, q_k - p_k)\}$, after which the verification process is terminated. All tokens generated up to this point are appended to the output sequence, and the speculation window is adjusted in preparation for the next iteration.

### 3.3. Entropy-based Probability Relaxation

In autoregressive (AR) image generation models, each token corresponds to an image patch. We observe that smooth regions exhibit high entropy and redundancy, making them easy for decoding. As illustrated in Figure 1, accelerating the generation of such high-entropy regions generally has limited effect on overall image quality. Therefore, entropy can be used as a criterion for relaxing probabilities in speculative decoding, enabling dynamic and adaptive acceleration. Additionally, entropy offers two further advantages: first, it is training-free, relying only on endogenous information produced during model inference, which allows for direct plug-and-play integration into any model; second, its computational overhead is low—unlike clustering-based methods that require sorting or pixel-based approaches that depend on VQ-VAE decoding—both of which can diminish the speed-up ratio.

However, probability relaxation may shift the output distribution, necessitating a corresponding resampling distribution to mitigate this effect. Let $b(x)$ denote a generalized acceptance probability defined over the vocabulary. The case where $b(x) > \min(1, \frac{q(x)}{p(x)})$ corresponds to the entropy-based relaxation described above. When $b(x) = \min(1, \frac{q(x)}{p(x)})$, the algorithm reduces to lossless speculative decoding. The scenario where $b(x) < \min(1, \frac{q(x)}{p(x)})$ is not considered, as lossless speculative decoding already provides better acceleration while maintaining exact distributional consistency.

Let the resampling distribution associated with $b(x)$ be denoted by $\mathcal{P}(x)$. The sequence distribution $P(x)$ generated by this approach can be expressed as:

$$P(x) = p(x)b(x) + \mathcal{P}(x)\sum_{x'}[1 - b(x')]p(x'). \quad (3)$$

Our goal is to minimize the discrepancy between $P(x)$ and the target distribution $q(x)$, measured by the total variation (TV) distance (Yin et al., 2024):

$$\text{Loss}(b) = \min_{\mathcal{P}} \text{TV}(P, q). \quad (4)$$

To provide a unified framework, we restrict our discussion to the case where $\min(1, \frac{q(x)}{p(x)}) \leq b(x) \leq 1$, having the following Theorem 3.1.

**Theorem 3.1.** *Let* $\min(1, \frac{q(x)}{p(x)}) \leq b(x) \leq 1$. *Then the resampling distribution defined by* $\mathcal{P}(x) = norm\{\max(0, q(x) - p(x))\}$ *achieves the minimum total variation loss, i.e.,* $Loss^*(b) = \min_{\mathcal{P}} TV(P, q) = \frac{1}{2}\sum_x |q(x) - b(x)p(x)| - \frac{1}{2}[1 - b(x)]p(x)$.

A proof of Theorem 3.1 is given in Appendix Section A. This result characterizes the optimal resampling distribution for any relaxed acceptance probability and provides an associated loss for control purposes.

### 3.4. Distribution Alignment Filter

Even under the optimal resampling distribution, relaxing acceptance probabilities inevitably introduces a deviation from the target distribution. The following theorem, adapted from (Yin et al., 2024), formalizes the relationship between the rejection probability and this distributional divergence.

**Theorem 3.2.** *Under the optimal resampling distribution, the following identity holds:*

$$P(reject) + \min_{\mathcal{P}} TV(P, q) = TV(p, q), \quad (5)$$

*where* $P(reject) = \sum_x[1 - b(x)]p(x)$

The proof of Theorem 3.2 is provided in Appendix Section B. This result reveals a fundamental trade-off. If the rejection probability is zero, then the divergence between the generated distribution and the target distribution remains exactly $\text{TV}(p, q)$. In this case, draft tokens are directly accepted in the final output. At the other extreme, reducing the total variation distance between $P$ and $q$ to zero requires the rejection probability to equal $\text{TV}(p, q)$. This latter case corresponds exactly to the behavior of lossless speculative decoding.

Based on Theorem 3.2, the TV distance provides a theoretical lower bound on the deviation between the generated distribution and the target one. To maintain generation quality, we design a dual acceptance criterion governed by the TV distance, which functions as a distributional alignment filter. When the TV distance is below a preset threshold, entropy-based probability relaxation is applied to accelerate sampling. If the TV distance exceeds the threshold, the accept criterion reverts to standard speculative decoding, thereby preserving exact distributional matching.

# 4. Experiments

## 4.1. Experiment Setups

**Benchmark.** We select the validation set of MS-COCO 2017 (Lin et al., 2014) as the benchmark dataset, which comprises natural scenes with complex backgrounds, diverse object layouts, and contextual relationships. Each image is accompanied by human-authored natural language descriptions, which serve as ground truth for image captioning generation tasks. For training-based methods, we follow the same setting as LANTERN (Jang et al., 2024), which choose ImageNet (Deng et al., 2009) and LAION-COCO (Schuhmann et al., 2022) as training dataset.

**Backbone Models.** We selected two recent and representative AR image generation models for experiments: Lumina-mGPT (Liu et al., 2024b) and Janus-Pro (Chen et al., 2025). For Lumina-mGPT, we utilized its 7B version to generate images with a resolution of $768 \times 768$ pixels. As for Janus-Pro, we employed its 1B and 7B version to generate images with a resolution of $384 \times 384$ pixels.

**Baselines.** We compare against two primary baselines: 1) Training-free methods, including Jacobi Decoding (Song et al., 2021), SJD (Teng et al., 2024), Amplify (Leviathan et al., 2023), Addition (Yin et al., 2024) and GSD (So et al., 2025); 2) Training-based methods, including EAGLE (Li et al., 2024) and LANTERN (Jang et al., 2024).

**Evaluation Metrics.** For visual quality, we use the CLIP Score (Radford et al., 2021) and the FID Score (Lin et al., 2014) as evaluation metrics. The former quantifies the semantic correspondence between generated images and text prompts, while the latter is computed by comparing the statistical distribution of generated images against that of the ground-truth validation set images. To evaluate generation speed, we measure the average latency and steps required to generate a single image.

**Implementation Details.** For Lumina-mGPT, we use top-k sampling with $k = 2000$ and classifier-free guidance (CFG) with a scale factor of 3. Following GSD (So et al., 2025), the temperature and window size are default set to 1 and 16, respectively. For Janus-Pro, we set the CFG scale to 5, which matches its default configuration.

## 4.2. Quantitative Comparison

We summarize the quantitative results for accelerating Lumina-mGPT and Janus-Pro in Table 1 and Table 2, respectively.

For Lumina-mGPT shown in Table 1, the proposed CSD achieves the highest CLIP score of 31.49 when accelerating by $2.6\times$, outperforming GSD by 0.16 in CLIP score with a reduction in inference time of 1.23s. This indicates that relaxing constraints on regions with high target entropy does not significantly compromise image quality. Additionally, our CSD achieves comparable results to training-based Eagle and Lantern methods in terms of both CLIP score and latency acceleration rate.

For Janus-Pro shown in Table 2, our CSD achieves the best performance in terms of latency and FID. Specifically, on the 7B model, at a comparable acceleration rate around $3.5\times$, our CSD with $\delta = 0.65$ outperforms the previous SOTA method GSD by 0.12 in CLIP score and 0.31 in FID, demonstrating substantial performance gains. Moreover, when prioritizing speed, relaxing the threshold $\delta$ to 0.8 enables CSD to reach $4.33\times$ latency acceleration, reducing inference time to 5.30s. This represents a 24% reduction compared to GSD and a 26% reduction compared to Addition, while maintaining competitive quality with a CLIP score of 32.21 and an FID of 33.58. On the 1B model, CSD remains equally effective, achieving up to $3.35\times$ latency reduction with competitive quality. Notably, the optimal $\delta$ shifts from 0.65 (7B) to 0.35 (1B), as smaller models exhibit greater discrepancy between target and draft distributions. This demonstrates that CSD is adaptive and robust: the entropy-based mechanism automatically responds to model capacity, and the $\delta$ threshold can be tuned to balance speed and quality across different model scales.

## 4.3. Qualitative Comparison

We further present the visualization results for image generation in Figure 3. In the top panel, our CSD generates complex and elaborate details in the face and clothing with the lowest latency, avoiding facial distortion or artifacts, compared to all methods except AR and JD. Although JD seems to achieve more nature face, it requires 20.32s almost consistent with the inference time of baseline AR. The bottom panel illustrates a surreal scene, where our method continues to properly preserve spatial relationships and retains fine-grained details. More visualization results are provided in Appendix Section C.

As show in Figure 4 we visualize the comparison of actual acceleration regions across different methods, where darker regions indicate that more tokens in the corresponding areas are predicted by the draft branch. It can be observed that previous methods fail to adaptively adjust acceleration regions according to image content. For instance, when generating a carrot, GSD still exhibits extensive dark acceleration regions on the main subject, while considerable acceleration potential remains unused in the plain background areas. In contrast, our proposed CSD concentrates acceleration on background regions, which explains its robustness under high compression ratios. Notably, since the carrot generation task is relatively simple, all methods achieve comparable CLIP scores, yet CSD attains the fastest inference speed of 6.37 s. For the more complex case of generating

*Table 1.* Quantitative results on MS-COCO benchmark with Lumina-mGPT(7B) as the baseline.

| METHOD | LATENCY($\downarrow$) | STEP($\downarrow$) | ACCELERATION RATE | | FID($\downarrow$) | CLIP SCORE ($\uparrow$) |
| | | | LATENCY($\uparrow$) | STEP($\uparrow$) | | |
| --- | --- | --- | --- | --- | --- | --- |
| LUMINA-MGPT 7B | | | | | | |
| VANILLA AR (LIU ET AL., 2024B) | 121.48s | 2379 | 1.00× | 1.00× | 30.90 | 31.31 |
| TRAINING-BASED | | | | | | |
| EAGLE (LI ET AL., 2024) | 57.85s | 809.2 | 2.10× | 2.94× | - | **33.3** |
| LANTERN (JANG ET AL., 2024) | **47.45s** | **655.4** | **2.56×** | **3.63×** | - | 32.7 |
| TRAINING-FREE | | | | | | |
| JD (SONG ET AL., 2021) | 117.91s | 2273.4 | 1.03× | 1.05× | **30.89** | 31.31 |
| SJD (TENG ET AL., 2024) | 53.73s | 1061.9 | 2.26× | 2.24× | 30.92 | 31.31 |
| AMPLIFY($k = 1.3$) (LEVIATHAN ET AL., 2023) | 47.85s | 921.1 | 2.54× | 2.58× | 33.03 | 31.24 |
| ADDITION($\epsilon = 0.1$) (YIN ET AL., 2024) | 48.25s | **909.3** | 2.52× | **2.62×** | 32.93 | 31.27 |
| GSD($G = 3$) (SO ET AL., 2025) | 47.99s | 925.2 | 2.53× | 2.57× | 31.61 | 31.33 |
| OURS($\lambda = 0.5, \delta = 0.35$) | **46.76s** | 922.8 | **2.60×** | 2.58× | 31.66 | **31.49** |

*Table 2.* Quantitative results on MS-COCO benchmark with Janus-Pro (1B & 7B) as the baseline.

| METHOD | LATENCY($\downarrow$) | STEP($\downarrow$) | ACCELERATION RATE | | FID($\downarrow$) | CLIP SCORE($\uparrow$) |
| | | | LATENCY($\uparrow$) | STEP($\uparrow$) | | |
| --- | --- | --- | --- | --- | --- | --- |
| JANUS-PRO 1B | | | | | | |
| VANILLA AR (LIU ET AL., 2024B) | 15.29s | 576 | 1.00× | 1.00× | 32.75 | 32.05 |
| SJD (TENG ET AL., 2024) | 7.86s | 303.8 | 1.95× | 1.90× | 32.49 | **32.05** |
| OURS($\lambda = 0.5, \delta = 0.35$) | 7.10s | 270.9 | 2.15× | 2.13× | 32.63 | 32.04 |
| OURS($\lambda = 0.5, \delta = 0.65$) | **4.56s** | **173.0** | **3.35×** | **3.55×** | **32.41** | 32.00 |
| JANUS-PRO 7B | | | | | | |
| VANILLA AR (LIU ET AL., 2024B) | 22.95s | 576 | 1.00× | 1.00× | 33.82 | 32.31 |
| JD (SONG ET AL., 2021) | 21.37s | 537.6 | 1.07× | 1.07× | 33.71 | **32.31** |
| SJD (TENG ET AL., 2024) | 11.64s | 298.8 | 1.97× | 1.93× | 33.85 | 32.28 |
| AMPLIFY($k = 4$) (LEVIATHAN ET AL., 2023) | 7.05s | 165.7 | 3.26× | 3.48× | 34.01 | 32.20 |
| ADDITION($\epsilon = 0.4$) (YIN ET AL., 2024) | 7.13s | 162.8 | 3.22× | 3.54× | 33.96 | 32.16 |
| GSD($G = 35$) (SO ET AL., 2025) | 6.93s | 161.7 | 3.31× | 3.56× | 34.00 | 32.16 |
| OURS($\lambda = 0.5, \delta = 0.65$) | 6.53s | 166.1 | 3.51× | 3.47× | 33.69 | 32.28 |
| OURS($\lambda = 0.5, \delta = 0.8$) | **5.30s** | **134.3** | **4.33×** | **4.29×** | **33.58** | 32.21 |

*Table 3.* Effects of different $\lambda$ and $\delta$

| $\lambda$ | $\delta$ | LATENCY($\downarrow$) | STEP($\downarrow$) | CLIP SCORE($\uparrow$) |
| --- | --- | --- | --- | --- |
| 0.5 | 0.65 | 6.53 | 166.1 | 32.28 |
| 0.3 | 0.65 | 6.96 | 172.6 | 32.28 |
| 0.7 | 0.65 | 6.46 | 155.8 | 32.25 |
| 0.5 | 0.45 | 9.44 | 223.5 | 32.31 |
| 0.5 | 0.75 | 6.02 | 145.9 | 32.25 |

*Table 4.* Effect of components

| CONFIGURATION | LATENCY($\downarrow$) | STEP($\downarrow$) | CLIP($\uparrow$) |
| --- | --- | --- | --- |
| W/O FILTER | | | |
| AMPLIFY($k = 1.3$) | 10.64 | 243.3 | 32.28 |
| ADDITION($\epsilon = 0.1$) | 10.64 | 252.3 | 32.27 |
| ENTROPY($\lambda = 0.02$) | 10.82 | 258.7 | 32.30 |
| W/ FILTER | | | |
| TV($\delta = 0.35$) | 9.69 | 252.7 | 32.32 |
| KL($\delta = 0.4$) | 10.36 | 258.0 | 32.32 |
| RKL($\delta = 0.4$) | 10.01 | 255.8 | 32.32 |

three dogs, CSD runs 2.8 s faster than SJD and achieves a 0.85-point improvement in CLIP score.

### 4.4. Ablation Study

**Effect of $\lambda$.** The optimal choice of $\lambda$ is stable and can be fixed without extensive tuning. As shown in Table 3, $\lambda = 0.5$ consistently yields a good trade-off between acceleration and quality, while larger $\lambda$ leads to noticeable performance degradation and smaller one decreases the acceleration rate with the same CLIP score. The relaxation coefficient $\lambda$ has a relatively minor effect on decoding speed, whereas large values of $\lambda$ lead to noticeable performance degradation. We therefore suggest $\lambda = 0.5$ as a suitable setting.

**Effect of $\delta$.** This hyperparameter is of critical importance, as the model architecture exerts significant influence on both performance and inference speed. As shown in Table 3, a larger $\delta$ yields high acceleration but at the expense of performance, whereas a smaller $\delta$ results in lower acceleration but

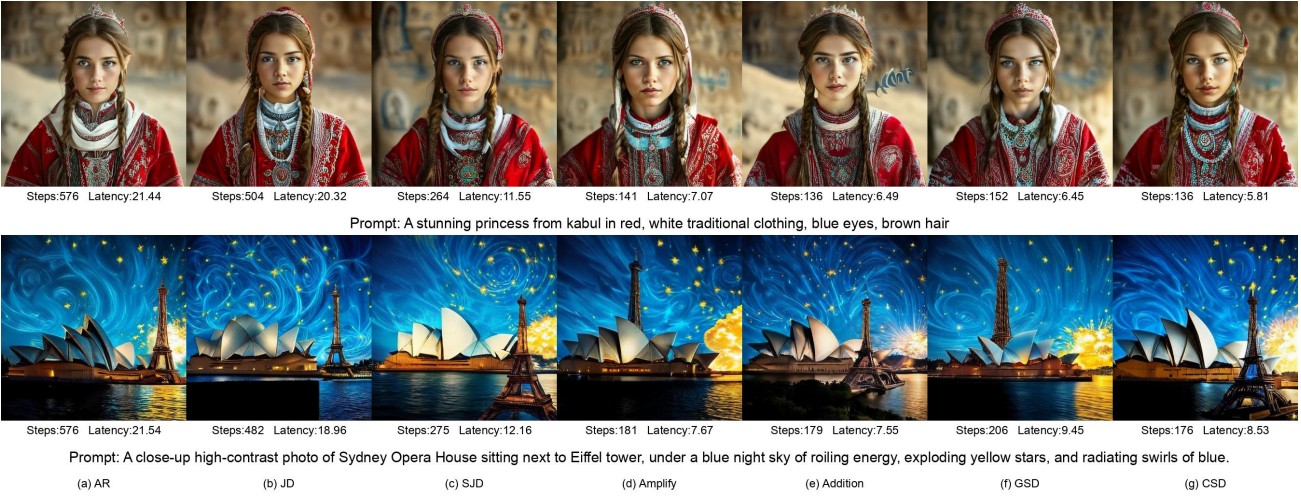

Prompt: A stunning princess from kabul in red, white traditional clothing, blue eyes, brown hair

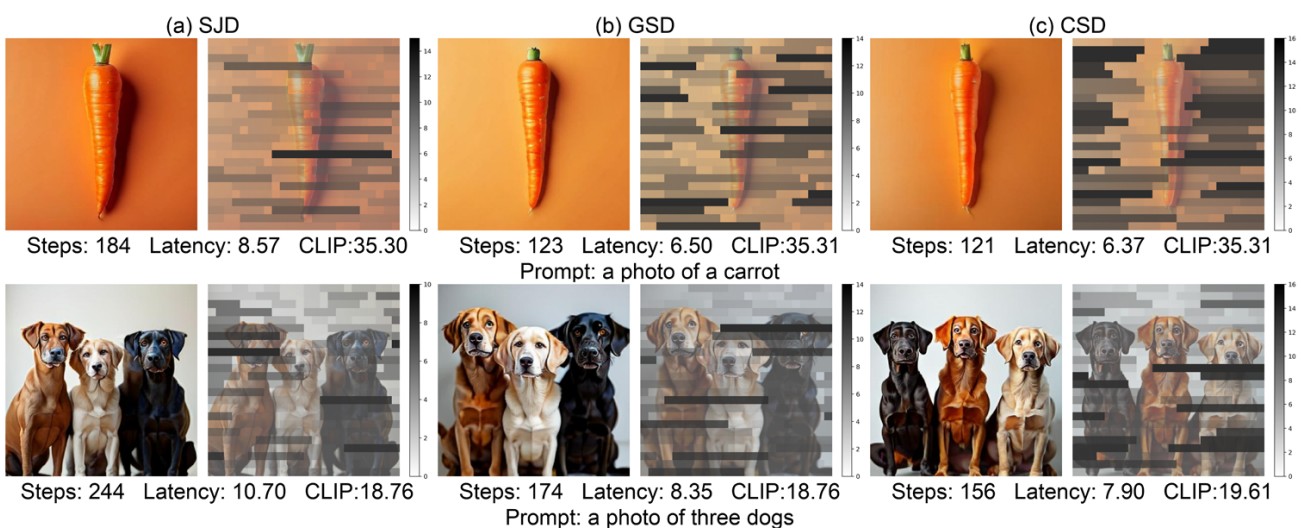

Prompt: A close-up high-contrast photo of Sydney Opera House sitting next to Eiffel tower, under a blue night sky of roiling energy, exploding yellow stars, and radiating swirls of blue.

(a) AR    (b) JD    (c) SJD    (d) Amplify    (e) Addition    (f) GSD    (g) CSD

*Figure 3.* Comparison between training-free methods on Janus-Pro(7B)

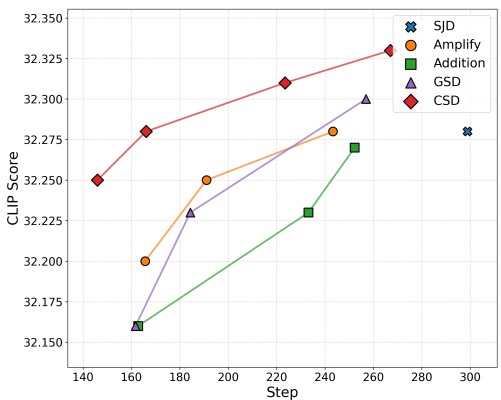

*Figure 4.* Visualization of accelerated tokens on Janus-Pro(7B)

*Figure 5.* Pareto-front comparison

superior performance stability. Consequently, we propose a search strategy starting with determining the target performance. For high performance, we aim for 0.35, and for high acceleration, 0.65. Then use a search with 0.1 interval based on results. Generally, tuning shouldn't exceed four times. Moreover, we also found that the optimal $\delta$ exhibits consistent behavior across different datasets. As shown in Table 5, the $\delta$ values validated on MS-COCO transfer well to the GenEval (Ghosh et al., 2023) benchmark. For the same Janus-Pro 7B model, setting $\delta = 0.65$ yields a $3.51\times$ acceleration on MS-COCO and a $3.36\times$ acceleration on GenEval, with both benchmarks exhibiting lossless visual quality.

**Effect of Individual Components.** In Table 4, we assess the impact of individual components. First, we evaluate three relaxation strategies without a filter—Amplify, Addi-

*Table 5.* Quantitative results on Geneval benchmark with Janus-Pro 7B as the baseline.

| METHOD | SINGLEOBJ. | TWOOBJ. | COUNTING | COLORS | POSITION | COLORATTRI. | OVERALL($\uparrow$) | LATENCY($\downarrow$) | STEPS($\downarrow$) |
|---|---|---|---|---|---|---|---|---|---|
| VANILLA AR | 0.88 | 0.97 | 0.59 | 0.64 | 0.78 | 0.88 | 0.79 | 22.95 | 576 |
| SJD (TENG ET AL., 2024) | 0.90 | 0.98 | 0.59 | 0.65 | 0.75 | 0.89 | 0.79 | 11.95 | 293.6 |
| OURS($\lambda = 0.5, \delta = 0.65$) | 0.88 | 0.97 | 0.60 | 0.63 | 0.77 | 0.88 | 0.79 | 6.84 | 161.7 |
| OURS($\lambda = 0.5, \delta = 0.8$) | 0.89 | 0.98 | 0.54 | 0.65 | 0.76 | 0.87 | 0.78 | 5.53 | 130.8 |

*Table 6.* Computational overhead comparison on Janus-Pro 7B using an MS-COCO 1000-subset.

| METHOD / COMPONENT | LATENCY (MS) |
|---|---|
| SJD (BASELINE) | 35.40 |
| GSD | 37.24 |
| CSD (OURS) | 35.46 |
|   -ENTROPY CALCULATION | 0.038 |
|   -TV CALCULATION & FILTERING | 0.020 |

tion, and Entropy—under comparable acceleration ratios. Among these, entropy-based relaxation achieves the highest CLIP score of 32.30, outperforming Amplify (32.28) and Addition (32.27), demonstrating its effectiveness as a guidance signal. Building upon the entropy-based relaxation strategy, we integrate a distribution alignment filter using three commonly used distribution discrepancy measures: Total Variation (TV), Kullback–Leibler (KL) divergence, and Reverse Kullback–Leibler (RKL) divergence. All three configurations yield nearly identical CLIP scores of 32.32. This indicates that adding a filter provides clear additional benefits, while the specific choice of discrepancy measure has only a marginal impact on the final performance. Taken together, both components play complementary roles: entropy-based relaxation accelerates inference in smooth regions, while the filter ensures that relaxation is only applied when draft and target distributions are well-aligned.

**Pareto Front Analysis.** In Figure 5, we extend the experimental evaluation by visualizing the pareto fronts, which illustrate the trade-off between acceleration ratio and CLIP score. Our method consistently achieves a favorable balance: it accelerates primarily smooth regions while leveraging the alignment filter to control distribution divergence, thereby preserving image quality. Notably, even at a high acceleration ratio (approximately 160 steps), our method attains a CLIP score of 32.28, remaining competitive with SJD (32.30). In contrast, other acceleration methods yield CLIP scores below 32.20 under the same conditions, indicating a sharper quality degradation. These results underscore the robustness of our approach in maintaining semantic alignment and perceptual quality across varying acceleration levels.

**Computational Overhead.** To evaluate the computational overhead of entropy calculation and different methods, we measured the average per-iteration latency on Janus-Pro 7B using an MS-COCO 1000-subset, as summarized in Table 6. The average per-iteration latency of CSD is 35.46 ms. Within this, the entropy calculation takes only 0.038 ms, and the TV calculation and filtering takes 0.020 ms. Compared to the SJD baseline, CSD adds only 0.06 ms per iteration—a mere 0.17% computation increase. Moreover, CSD achieves lower computation overhead than GSD (35.46 ms vs. 37.24 ms), as GSD relies on clustering-based grouping that requires sorting and distance computations. Thus, CSD preserves speculative decoding efficiency with negligible extra cost.

## 5. Conclusion

This work tackles the inefficiency of autoregressive (AR) image generation and the content-agnostic flaw of existing speculative decoding (SD) methods. We propose Content-aware Speculative Decoding (CSD), a novel SD algorithm that leverages the target model's entropy as a content signal to enable adaptive, high-quality acceleration for AR image generation. CSD integrates an entropy-based probabilistic relaxation mechanism with an optimal resampling strategy, dynamically tuning token acceptance probabilities by image regions' informational uncertainty. Moreover, its total variation-based distribution alignment filter aligns generated token distributions with the target model. Extensive experiments on Lumina-mGPT and Janus-Pro validate CSD's superiority over SOTA SD methods.

## Acknowledgements

This work is supported by the National Natural Science Foundation of China (NO. 62572193, NO. 12371289), China Postdoctoral Science Foundation (NO. 2024M760930), the Open Research Fund of the Key Laboratory of Advanced Theory and Application in Statistics and Data Science, Ministry of Education, and the Fundamental Research Funds for the Central Universities.

## Impact Statement

This paper presents work whose goal is to advance the field of Machine Learning. There are many potential societal consequences of our work, none which we feel must be specifically highlighted here.

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

# Appendix

**Summary.** This appendix provides proofs of key theorems and additional visual results. Appendix Section A presents the proof of Theorem 3.1, Appendix Section B provides the proof of Theorem 3.2 and Appendix Section C offers more visualized results and analysis on Janus-Pro.

## A. Proof of Theorem 3.1

**Theorem 3.1.** *Let* $\min(1, \frac{q(x)}{p(x)}) \leq b(x) \leq 1$. *Then the resampling distribution defined by* $\mathcal{P}(x) = norm\{\max(0, q(x) - p(x))\}$ *achieves the minimum total variation loss, i.e., Loss*$^*(b) = \min_{\mathcal{P}} TV(P, q) = \frac{1}{2}\sum_x |q(x) - b(x)p(x)| - \frac{1}{2}[1 - b(x)]p(x)$.

*Proof.* Define $A(x) = \dfrac{q(x) - b(x)p(x)}{\sum_{x'}[1 - b(x')]p(\hat{x})}$ and the sets $A_+ = \{x : A(x) \geq 0\}$, $A_- = \{x : A(x) < 0\}$. The optimal resample distributions are $\{\mathcal{P}^* : \mathcal{P}^*|_{A_-}(\cdot) = 0; 0 \leq \mathcal{P}^*|_{A_+}(\cdot) \leq A(x)\}$, and the corresponding discrepancy is Loss$^*(b) = \frac{1}{2}\sum_x |q(x) - b(x)p(x)| - \frac{1}{2}[1 - b(x)]p(x)$.

First, we prove that for any optimal solution $\mathcal{P}^*$, it must satisfy $\mathcal{P}^*(x) = 0$ for all $x \in A_-$.

Assume for contradiction that there exists $\bar{x} \in A_-$ such that $\mathcal{P}^*(x) > 0$.

Suppose that for all $x \in A_+$, we have $A(x) \leq \mathcal{P}^*(x)$.

Then

$$1 = \sum_x A(x) = \sum_{x \in A_-} A(x) + \sum_{x \in A_+} A(x) \leq A(\bar{x}) + \sum_{x \in A_+} A(x)$$
$$< \sum_{x \in A_+} A(x) \leq \sum_{x \in A_+} \mathcal{P}^*(x) \leq \sum_x \mathcal{P}^*(x) = 1$$

which is a contradiction. Thus, there exists $\hat{x} \in A_+$ such that $A(\hat{x}) > \mathcal{P}^*(\hat{x})$.

We have $-\mathcal{P}^*(\bar{x}) < 0$ and $A(\hat{x}) - \mathcal{P}^*(\hat{x}) > 0$. By the triangle inequality,

$$|-\mathcal{P}^*(\bar{x})| + |A(\hat{x}) - \mathcal{P}^*(\hat{x})| > |A(\hat{x}) - \mathcal{P}^*(\hat{x}) - \mathcal{P}^*(\bar{x})|.$$

Since $A(\bar{x}) < 0$,

$$|A(\bar{x})| + |-\mathcal{P}^*(\bar{x})| = |A(\bar{x}) - \mathcal{P}^*(\bar{x})|.$$

Adding $|A(\bar{x})|$ to both sides yields

$$|A(\bar{x}) - \mathcal{P}^*(\bar{x})| + |A(\hat{x}) - \mathcal{P}^*(\hat{x})| > |A(\bar{x})| + |A(\hat{x}) - \mathcal{P}^*(\hat{x}) - \mathcal{P}^*(\bar{x})|.$$

Now, construct a new distribution $\mathcal{P}'(x)$ as follows:

$$\mathcal{P}'(x) = \begin{cases} \mathcal{P}^*(x), & x \notin \{\bar{x}, \hat{x}\}, \\ 0, & x = \bar{x}, \\ \mathcal{P}^*(x) + \mathcal{P}^*(\bar{x}), & x = \hat{x}. \end{cases}$$

Substituting the optimal solution into the objective function, we have

$$
\begin{aligned}
\sum_x |A(x) - \mathcal{P}^*(x)| &= \sum_{x \notin \{\bar{x}, \hat{x}\}} |A(x) - \mathcal{P}^*(x)| + |A(\bar{x}) - \mathcal{P}^*(\bar{x})| + |A(\hat{x}) - \mathcal{P}^*(\hat{x})| \\
&= \sum_{x \notin \{\bar{x}, \hat{x}\}} |A(x) - \mathcal{P}'(x)| + |A(\bar{x}) - \mathcal{P}^*(\bar{x})| + |A(\hat{x}) - \mathcal{P}^*(\hat{x})| \\
&> \sum_{x \notin \{\bar{x}, \hat{x}\}} |A(x) - \mathcal{P}'(x)| + |A(\bar{x})| + |A(\hat{x}) - \mathcal{P}^*(\hat{x}) - \mathcal{P}^*(\bar{x})| \\
&= \sum_{x \notin \{\bar{x}, \hat{x}\}} |A(x) - \mathcal{P}'(x)| + |A(\bar{x}) - \mathcal{P}'(\bar{x})| + |A(\hat{x}) - \mathcal{P}'(\hat{x})| \\
&= \sum_x |A(x) - \mathcal{P}'(x)|.
\end{aligned}
$$

This implies that $\mathcal{P}'(x)$ yields a better objective function value, contradicting the optimality of $\mathcal{P}^*(x)$.

Therefore, $\mathcal{P}^*(x) = 0$ for all $\bar{x} \in A_-$.

Next, we compute the optimal objective function value:

$$
\begin{aligned}
\sum_x |A(x) - \mathcal{P}^*(x)| &= \sum_{x \in A_-} |A(x) - \mathcal{P}^*(x)| + \sum_{x \in A_+} |A(x) - \mathcal{P}^*(x)| \\
&= \sum_{x \in A_-} |A(x)| + \sum_{x \in A_+} |A(x) - \mathcal{P}^*(x)|.
\end{aligned}
$$

Consider the generalization of the triangle inequality to $n$ terms:

$$
|\sum_{x=1}^n a_x| \le \sum_{x=1}^n |a_x|.
$$

Applying this, we have

$$
\begin{aligned}
\sum_x |A(x) - \mathcal{P}^*(x)| &\ge \sum_{x \in A_-} |A(x)| + |\sum_{x \in A_+} A(x) - \sum_{x \in A_+} \mathcal{P}^*(x)| \\
&= \sum_{x \in A_-} |A(x)| + |\sum_{x \in A_+} A(x) - 1| \\
&= \sum_{x \in A_-} |A(x)| + \sum_{x \in A_+} A(x) - 1 \\
&= \sum_x |A(x)| - 1,
\end{aligned}
$$

with equality if and only if $A(x) - \mathcal{P}^*(x) \ge 0$ for all $x \in A_+$

Substituting into the original objective function:

$$
\frac{R}{2} \sum_x |A(x) - \mathcal{P}(x)| = \frac{R}{2} \sum_x |A(x)| - \frac{R}{2} = \frac{1}{2} \sum_x |q(x) - b(x)p(x)| - \frac{1}{2} \sum_x [(1 - b(x))p(x)].
$$

In summary, the set of optimal solutions is

$$
\{\mathcal{P}^* : \mathcal{P}^*|_{A_-}(\cdot) = 0; 0 \le \mathcal{P}^*|_{A_+}(\cdot) \le A(\cdot)\},
$$

and the optimal objective function value is

$$\frac{1}{2}\sum_x |q(x) - b(x)p(x)| - \frac{1}{2}\sum_x [(1 - b(x))p(x)].$$

We define

$$D(x) = q(x) - b(x)p(x), \quad R = \sum_x [1 - b(x)]p(x), \quad A(x) = \frac{D(x)}{R},$$

$$A_+ = \{x : A(x) \geq 0\} = \{x : D(x) \geq 0\}, \quad A_- = \{x : A(x) < 0\} = \{x : D(x) < 0\}.$$

Using the positive part of $D(x) > 0$, we construct a distribution $\mathcal{P}_{norm}(x)$:

$$D_+(x) = \max(0, D(x)), \quad S = \sum_x D_+(x) = \sum_{x \in A_+} D(x).$$

$$\mathcal{P}_{norm}(x) = \frac{D_+(x)}{S} = \begin{cases} \dfrac{D(x)}{S}, & x \in A_+, \\ 0, & x \in A_-. \end{cases}$$

This distribution clearly satisfies non-negativity and normalization, i.e., $\mathcal{P}_{norm}(x) \geq 0$ and $\sum_x \mathcal{P}_{norm}(x) = 1$.
By definition,

$$\mathcal{P}_{norm}|_{A_-}(\cdot) = 0.$$

Moreover,

$$\begin{aligned} R &= \sum_x [1 - b(x)]p(x) \\ &= \sum_x p(x) - \sum_x b(x)p(x) \\ &= 1 - \sum_x b(x)p(x) \\ &= \sum_x q(x) - \sum_x b(x)p(x) \\ &= \sum_x [q(x) - b(x)p(x)] \\ &= \sum_x D(x). \end{aligned}$$

Thus,

$$R = \sum_x D(x) = \sum_{x \in A_+} D(x) + \sum_{x \in A_-} D(x) = S + \sum_{x \in A_-} D(x).$$

For $x \in A_-$, we have $D(x) < 0$, so $\sum_{x \in A_-} D(x) \leq 0$. Therefore,

$$R = S + \sum_{x \in A_-} D(x) \leq S.$$

For $x \in A_+$,

$$\mathcal{P}_{norm}(x) = \frac{D(x)}{S} \leq \frac{D(x)}{R} = A(x).$$

Hence,

$$\mathcal{P}_{norm}(x) \in \{\mathcal{P}^* : \mathcal{P}^*|_{A_-}(\cdot) = 0; 0 \leq \mathcal{P}^*|_{A_+}(\cdot) \leq A(\cdot)\}.$$

If $b(x) = \min(1, \frac{q(x)}{p(x)})$, then

$$D(x) = q(x) - min(1, \frac{q(x)}{p(x)}) \cdot p(x) = q(x) - min(p(x), q(x)) = \max(0, q(x) - p(x)) > 0,$$

and

$$\mathcal{P}_{norm}(x) = \text{norm}\{\max(0, q(x) - p(x))\},$$

which aligns with standard speculative decoding.

If we relax the acceptance probability, i.e., $b(x) = \min(1, \frac{q(x)}{p(x)} + \epsilon)$ with $\epsilon > 0$, then

$$D(x) = q(x) - min(1, \frac{q(x)}{p(x)} + \epsilon) \cdot p(x) = q(x) - min(p(x), q(x) + \epsilon p(x)) = \max(-\epsilon p(x), q(x) - p(x)),$$

and

$$\mathcal{P}_{norm}(x) = \text{norm}\{\max(0, q(x) - p(x))\}$$

Therefore, regardless of how much the acceptance probability is relaxed, the optimal resampling distribution remains $\text{norm}\{\max(0, q(x) - p(x))\}$. $\square$

## B. Proof of Theorem 3.2

**Theorem 3.2.** *Under the optimal resampling distribution, the following identity holds:* $P(reject) + \min_{\mathcal{P}} TV(P, q) = TV(p, q)$, *where* $P(reject) = \sum_x [1 - b(x)]p(x)$

*Proof.* Given

$$\text{Loss}^*(b) = \frac{1}{2}\sum_x |q(x) - b(x)p(x)| - \frac{1}{2}\sum_x [1 - b(x)]p(x),$$

we have

$$\text{Loss}^*(b) + P(\text{reject}) = \frac{1}{2}\sum_x |q(x) - b(x)p(x)| - \frac{1}{2}\sum_x [1 - b(x)]p(x) + \sum_x [1 - b(x)]p(x)$$

$$= \frac{1}{2}\sum_x |q(x) - b(x)p(x)| + \frac{1}{2}\sum_x [1 - b(x)]p(x).$$

To prove the desired result,

$$\text{Loss}^*(b) + P(\text{reject}) = \text{TV}(p, q) = \frac{1}{2}\sum_x |p(x) - q(x)|,$$

it suffices to show that

$$\sum_x |q(x) - b(x)p(x)| + \sum_x [1 - b(x)]p(x) = \sum_x |p(x) - q(x)|.$$

Since $\min(1, \frac{q(x)}{p(x)}) \le b(x) \le 1$, then we prove the following stronger claim

$$|q(x) - b(x)p(x)| + (1 - b(x))p(x) = |p(x) - q(x)|.$$

- If $q(x) \ge p(x)$, then $1 = \min(1, \frac{q(x)}{p(x)}) \le b(x) \le 1$ implies $b(x) = 1$, so the above is equivalent to $|q(x) - p(x)| = |p(x) - q(x)|$ is always true;

- If $q(x) < p(x)$, then $b(x)p(x) \ge \min(p(x), q(x)) = q(x)$. In this case

$$|q(x) - b(x)p(x)| + (1 - b(x))p(x) = b(x)p(x) - q(x) + (1 - b(x))p(x) = p(x) - q(x) = |p(x) - q(x)|.$$

This concludes the proof. $\square$

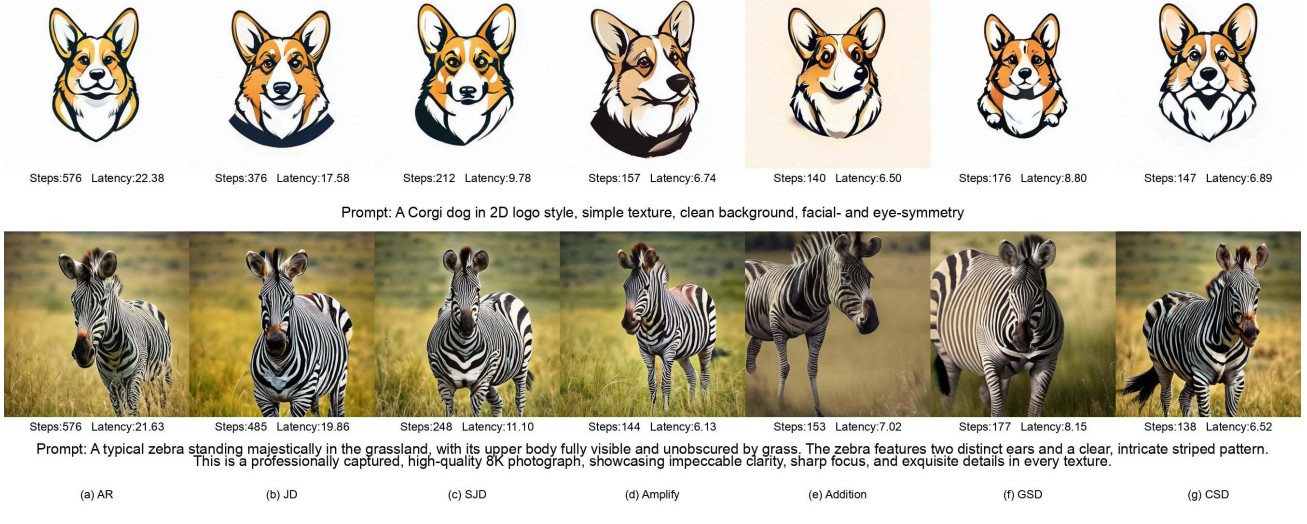

| Steps:576 Latency:22.38 | Steps:376 Latency:17.58 | Steps:212 Latency:9.78 | Steps:157 Latency:6.74 | Steps:140 Latency:6.50 | Steps:176 Latency:8.80 | Steps:147 Latency:6.89 |

Prompt: A Corgi dog in 2D logo style, simple texture, clean background, facial- and eye-symmetry

| Steps:576 Latency:21.63 | Steps:485 Latency:19.86 | Steps:248 Latency:11.10 | Steps:144 Latency:6.13 | Steps:153 Latency:7.02 | Steps:177 Latency:8.15 | Steps:138 Latency:6.52 |

Prompt: A typical zebra standing majestically in the grassland, with its upper body fully visible and unobscured by grass, The zebra features two distinct ears and a clear, intricate striped pattern. This is a professionally captured, high-quality 8K photograph, showcasing impeccable clarity, sharp focus, and exquisite details in every texture.

| (a) AR | (b) JD | (c) SJD | (d) Amplify | (e) Addition | (f) GSD | (g) CSD |

*Figure 6.* More visualization on Janus-Pro

## C. More Visualization

Figure 6 provides further visualization results on Janus-Pro. The first row highlights that SJD, Amplify, and Addition fail to strictly adhere to the textual instructions, suffering from issues such as structural asymmetry and inconsistent background colors. In contrast, our method effectively identifies and preserves key compositional elements, ensuring superior instruction fidelity. Furthermore, as illustrated in the second row, both Addition and GSD introduce significant deviations from the reference image in background regions. Our approach, however, maintains high background consistency via the distribution alignment filter, even in these non-salient areas.

