# OpenReview forum: "CSD: Content-aware Speculative Decoding for Efficient Image Generation"
_ICML.cc/2026/Conference — ICML 2026 regular_

### Official Review · Reviewer_9GJz · 2026-03-08

**Soundness:** 3
**Presentation:** 2
**Significance:** 3
**Originality:** 3
**Overall Recommendation:** 4
**Confidence:** 3

**Summary:**

This paper proposes a content-aware speculative decoding algorithm (CSD) to accelerate autoregressive image generation. The method observes that natural images contain heterogeneous regions and leverages target model entropy to relax token acceptance in smoother, low-detail areas. A distribution alignment filter based on total variation distance is also introduced to prevent excessive divergence from the target distribution to maintain visual quality. The authors provide theoretical analysis soundly supporting the proposed algorithm. Experiments demonstrate significant inference speedups with minimal degradation in generation quality.

**Compliance With Llm Reviewing Policy:**

Affirmed.

**Final Justification:**

I update the score to 4.
After reading the authors’ rebuttal, I find that my main concerns have been addressed. The clarification of the overall framework and the motivation for why the proposed method is especially relevant for image generation have improved my assessment of the work.

**Key Questions For Authors:**

1. The paper introduces relaxation-based techniques and motivates why these heuristics are necessary to complement the original speculative decoding. However, the experiments do not seem to include a comparison with the conventional speculative decoding applied to images. This omission could be a significant weakness of this work.

2. This method sacrifices losslessness to improve speed, yet in some cases, the results show better FID and CLIP scores than other baselines. How can this phenomenon be explained?

3. The proposed algorithm seems quite general, and the content-aware mechanism basically means relying on entropy, which is not strictly image-specific. This naturally raises the question: if it were applied to a simple text dataset for comparison with standard speculative decoding, what results would we obtain? If there are differences compared to images, analyzing why images behave differently could be a fruitful direction for research.

4. What happens if we change the relaxation strategy for low-entropy versus high-entropy regions? Would the results differ significantly? A deeper analysis on how to choose regions for relaxation based on entropy could provide valuable insights.

5. How sensitive is the method to model size, and would the results hold if smaller or larger models are used?

6. How much of the performance improvement comes from the entropy-based relaxation versus the alignment filter? Could there be scenarios where the filter dominates and relaxation provides little benefit?

**Limitations:**

The authors state: "There are many potential societal consequences of our work." However, they do not specify what these consequences are. The authors should provide a concrete description of the potential impacts.

**Strengths And Weaknesses:**

Strengths

1. The paper presents a good adaptation of speculative decoding to autoregressive image generation, achieving inference speedup.

2. The theoretical analysis is sound and provides justification for the proposed relaxation and resampling mechanisms.

3. The introduction of a distribution alignment filter serves as an effective safety mechanism to control distributional divergence and preserve generation quality.

Weaknesses

1. The experiments are conducted only on MS-COCO, which limits the evaluation diversity. Additional benchmarks with more complex scenes or higher-resolution images would strengthen the generalizability claims.

2. The paper does not clearly describe the speculative decoding setup, particularly the small draft model (I guess the mentioned 7B models are base or target models, but it is hard to tell reading this paper). The architecture, size, and training cost of the auxiliary model are unspecified, making reproducibility and practical deployment unclear.

3. The speedup over strong training-free baselines is relatively modest.

---

> ### Author Rebuttal · Authors · 2026-03-31
>
> ### W1. Benchmark Diversity:
> Our experiments already cover two resolutions (384x384 and 768x768) across two autoregressive models. To further address diversity, we provide additional GenEval results, where CSD achieves comparable overall scores (0.79) to Janus-Pro 7B and SJD, with 3.36x latency acceleration at δ=0.65. Moreover, CSD accelerates Janus-Pro 7B by 4.15x with a negligible performance drop of 0.01, significantly outperforming SJD’s 1.96x.
> ### W2. Clarity of Speculative Decoding Setup:
> We clarify that our method **does not employ a 7B target model as the small draft one, and our source codes in the supplementary materials provide the evidence**. Following SJD, we adopt Jacobi Decoding to generate draft tokens **without requiring any auxiliary model training**. This makes our approach **training-free and plug-and-play**, as stated in Section 3.3.
> ### W3. Speedup Over Strong Training-Free Baselines:
> On Janus-Pro with δ=0.8, CSD achieves 4.33x latency acceleration (5.30s) while maintaining competitive quality (CLIP 32.21, FID 33.58). CSD reduces inference time by 24% and 26% compared to GSD (6.93s) and Addition (7.13s), with the best CLIP/FID scores among all baselines. More details refer to W2 of 9B7J.
> ### Q1. Comparison with Conventional SD:
> We agree that conventional SD has proven effective in text generation. However, its application to autoregressive image generation faces the following challenges:
>
> - **Practical compatibility constraints**: Models like Janus-Pro 1B lack publicly available smaller draft models that share the same visual codebook, rendering vanilla SD inapplicable.
>
> - **Domain-specific low acceptance rates**: As noted by LANTERN and shown in their Figure 2, SD in vision tasks suffers from **significantly lower acceptance rates** compared to text, due to the multimodal nature of visual tokens. Consequently, subsequent works-including SJD, GSD, and LANTERN itself-have all moved the comparison of vanilla SD, and proposed specialized techniques tailored for image generation.
>
> - **Theoretical analysis**: Following the expected speedup formula in our paper (Eq. 2):$\frac{1-\alpha^{\gamma+1}}{(1-\alpha)(\gamma c+1)}$, where $c$ is the draft-target runtime ratio. For a realistic setup with Janus-Pro (1B/7B), we have $c\approx0.8$, the critical acceptance rate $\alpha$ required to achieve speedup > 1 is 0.800 for $\gamma=1$, 0.860 for $\gamma=2$ and 0.927 for $\gamma=5$—**far above the acceptance rates typically observed in visual token generation**.
>
> ### Q2. Explanation of the Improved Metric:
> CSD may improve FID/CLIP despite relaxing distribution matching due to: (1) CLIP robustness: GSD shows CLIP remains high even with 50% token shuffling; (2) visual redundancy: multiple token sequences yield perceptually similar images, and relaxation in smooth regions may select equally valid or better alternatives.
> ### Q3. Applicability to Text:
> Entropy in image generation differs from text: smooth backgrounds show high entropy but are easy, while fine details show low entropy but are hard. This reversed correlation makes entropy-based relaxation effective for images but potentially harmful for text, where high entropy typically indicates genuine difficulty[1].
>
> [1] Scalable best-of-n selection for large language models via self-certainty. arXiv 2025.
> ### Q4. Impact of Changing Relaxation Strategy:
> To further validate the effectiveness of our entropy-based relaxation strategy, we extend the ablation study in Table 4 by including two additional variants: **Entropy** (our original strategy) and **Inverse Entropy** (supported by the conclusion of Q3).
>
> For clarity, we define the Inverse Entropy strategy as:
> $\epsilon=\lambda(H_{max}-H(q_k))$, where $H_{max}=log|V|$ and $|V|=16384$ for Janus-Pro. The results reveal that **entropy-based relaxation achieves the highest CLIP score (32.30)** among filter-free strategies. In contrast, **inverse entropy yields a CLIP score of 32.27**, which is consistent to Addition. This finding suggests that simply reversing the relaxation strategy **does not yield meaningful improvements** over constant relaxation.
>
> | Configuration | Latency($\downarrow$) | Step($\downarrow$) | CLIP($\uparrow$) |
> |---------------|------------|---------|---------|
> | Entropy | 10.82    | 258.7   | 32.30   |
> | Inverse Entropy | 10.74 | 254.5 | 32.27 |
>
> ### Q5. Sensitivity to Model Size:
> On Janus-Pro 1B, CSD achieves 3.35x latency reduction with competitive quality (CLIP 32.00 vs. 32.05). Optimal δ shifts from 0.65 (7B) to 0.35 (1B), showing CSD adapts to model capacity. More details refer to W1 of WdW4.
> ### Q6. Contribution of Entropy-based Relaxation vs. Alignment Filter:
> As shown in Q4, entropy-based relaxation alone improves CLIP from 32.27 (Addition) to 32.30; adding the TV filter further raises it to 32.32. The two components are complementary: relaxation accelerates smooth regions, while the filter ensures it applies only when distributions are aligned, preventing quality degradation.

---

> > ### Author Rebuttal · Reviewer_9GJz · 2026-04-01
> >
> > After reading the authors’ rebuttal, I find that my main concerns have been addressed.

---

> > > ### Author Response · Authors · 2026-04-02
> > >
> > > Dear Reviewer 9GJz,
> > >
> > > Thank you again for your invaluable time and the effort on our paper. Thank you very much for approving our work!
> > >
> > > Sincerely Yours,
> > >
> > > The Authors

---

### Official Review · Reviewer_WdW4 · 2026-03-10

**Soundness:** 2
**Presentation:** 2
**Significance:** 2
**Originality:** 2
**Overall Recommendation:** 3
**Confidence:** 3

**Summary:**

This paper proposes a content-aware speculative decoding algorithm aimed at accelerating the inference of autoregressive (AR) image generation models. The core idea is to use the entropy of the target model as an indicator of uncertainty. The algorithm relaxes the acceptance criteria of speculative decoding in smooth / low-detail regions of the image, and introduces a Distribution Alignment Filter to mitigate potential degradation in generation quality.

**Compliance With Llm Reviewing Policy:**

Affirmed.

**Key Questions For Authors:**

See the weaknesses above.

**Limitations:**

yes

**Strengths And Weaknesses:**

Strengths:
1. Introducing entropy into speculative decoding aligns well with the image generation process of AR architectures.
2. The method achieves a 3.51× acceleration on models such as Janus-Pro, demonstrating significant performance improvement and offering practical engineering value.

Weaknesses:
1. The theoretical threshold $\delta$ used in the distribution alignment filter lacks general guidance. Experiments show that the optimal value is $\delta=0.65$ for Janus-Pro, while Lumina-mGPT requires $\delta=0.35$. It is unclear why the optimal $\delta$ differs so significantly between the two models.
2. The description in Section 3.3 fundamentally contradicts the experimental observations shown in Figure 1. The paper states that smooth regions have “low uncertainty and are easy to predict,” yet in the visualization these regions are labeled as “high-entropy regions.”
3. The paper categorizes MaskGIT (Chang et al., 2022) as a method aimed at accelerating autoregressive (AR) image generation. The authors argue that existing approaches accelerate AR models by modifying the architecture and present CSD (training-adaptive / training-free) as a contrasting approach. However, MaskGIT is a non-autoregressive (NAR) model, and its parallel generation capability stems from its inherent design paradigm rather than being an architectural modification intended to accelerate AR models.

---

> ### Author Rebuttal · Authors · 2026-03-31
>
> We sincerely thank the reviewers for recognizing our introduction of entropy, the significant performance improvement and practical engineering value. Below we response the concerns point-to-point:
>
> ### W1. Lack of General Guidance for Threshold $\delta$
> Thanks for your insightful observation. We assume that the difference regarding the optimal $\delta$ across different models stems from **inherent discrepancies in the target models' output distributions**. In addition, our design principle is to maintain CLIP score relatively consistent with vanilla AR models, while exploring the maximum potential for model acceleration. Thus, the reported Threshold $\delta$ is different.
>
> - **Janus-Pro ($\delta$=0.65):** This model employs a better training strategy with larger-scale data and optimized multimodal alignment, resulting in a **closer alignment between its draft and target distributions**. Consequently, a larger $\delta$ can be applied without compromising quality.
>
> - **Lumina-mGPT ($\delta$=0.35):** This model exhibits **greater divergence between the draft and target distributions**, requiring a stricter $\delta$ to preserve fidelity.
>
> This suggests that **$\delta$ reflects the model's intrinsic distributional stability** rather than an arbitrary heuristic.
>
> We also add the new experiments to evaluate the sensitivity of threshold $\delta$ on different model sizes and benchmark datasets. We found that **the optimal value depends on model capacity, but exhibits consistent transferability across datasets.**
>
> For larger or higher-capacity models (e.g., Janus-Pro-7B), a larger $\delta$ (e.g., 0.65) can be adopted without sacrificing quality, as the model is more robust to distributional shifts. For smaller models (e.g., Janus-Pro-1B), a conservative $\delta$ (e.g., 0.35) preserves semantic consistency.
>
> Consequently, we propose a search strategy starting with determining the target performance. For high performance, we aim for 0.35, and for high acceleration, 0.65. Then use a search with 0.1 interval based on results. Generally, tuning shouldn't exceed four times.
>
> Moreover, we also found that the optimal $\delta$ exhibits consistent behavior across different datasets. For instance, the $\delta$ values validated on MS-COCO transfer well to the GenEval benchmark, suggesting that the content-aware mechanism is not overly sensitive to dataset shifts.
>
> The above results and analysis will be added to our revision.
>
> | Method | Latency($\downarrow$) | Step($\downarrow$) | FID($\downarrow$) | CLIP score($\uparrow$) |
> | :--- | :--- | :--- | :--- | :--- |
> | Janus-Pro 1B |  |  |  |  |
> | Vanilla AR | 15.29s (1.00x) | 576 (1.00x) | 32.75 | 32.05 |
> | SJD  | 7.86s (1.95x) | 303.8 (1.90x) | 32.49 | 32.05 |
> | Ours($\lambda=0.5, \delta=0.35$) | 7.10s (2.15x) | 270.9 (2.13x) | 32.63 | 32.04 |
> | Ours($\lambda=0.5, \delta=0.65$) | 4.56s (3.35x) | 173.0 (3.55x) | 32.41 | 32.00 |
> | Janus-Pro 7B |  |  |  |  |
> | Vanilla AR | 22.95s (1.00x) | 576 (1.00x) | 33.82 | 32.31 |
> | JD | 21.37s (1.07x) | 537.6 (1.07x) | 33.71 | 32.31 |
> | SJD | 11.64s (1.97x) | 298.8 (1.93x) | 33.85 | 32.28 |
> | Amplify($k=4$) | 7.05s (3.26x) | 165.7 (3.48x) | 34.01 | 32.20 |
> | Addition($\epsilon=0.4$) | 7.13s (3.22x) | 162.8 (3.54x) | 33.96 | 32.16 |
> | GSD($G=35$) | 6.93s (3.31x) | 161.7 (3.56x) | 34.00 | 32.16 |
> | Ours($\lambda=0.5,\delta=0.65$) | 6.53s (3.51x) | 166.1 (3.47x) | 33.69 | 32.28 |
> | Ours($\lambda=0.5, \delta=0.8$) | 5.30s (4.33x) | 134.3 (4.29x) | 33.58 | 32.21 |
>
> | Method | Latency($\downarrow$) | Step($\downarrow$) | Overall($\uparrow$) |
> | :--- | :--- | :--- | :--- |
> | Vanilla AR (Janus-Pro 7B) | 22.95s (1.00x) | 576 (1.00x) | 0.79 |
> | SJD | 11.95s (1.92x) | 293.6 (1.96x) | 0.79 |
> | Ours($\lambda=0.5, \delta=0.65$) | 6.84s (3.36x) | 161.7 (3.56x) | 0.79 |
> | Ours($\lambda=0.5, \delta=0.8$) | 5.53s (4.15x) | 130.8 (4.40x) | 0.78 |
>
> ### W2. Contradiction Between Section 3.3 and Figure 1
> Sorry for this typo and confusion. The correct conclusion is that smooth regions exhibit high entropy and redundancy, making them easy for decoding. The experimental observations and all other discussions are consistent with this corrected interpretation. We will correct this typo in Section 3.3.
>
>
> ### W3. Mischaracterization of MaskGIT
>
> We are sorry for this incorrect description of MaskGIT. To avoid conceptual confusion and accurately position the contributions of our work, we have revised the Introduction as follows:
>
> > In contrast to the sequential nature of autoregressive models, non-autoregressive (NAR) approaches ( e.g., MaskGIT (Chang et al., 2022) and AutoNAT (Ni et al., 2024a)) aim to generate images via parallel decoding and mask prediction. However, their parallel generation capability stems from their inherent design paradigm and cannot be directly transferred to autoregressive generation without fundamentally altering the model architecture.

---

> > ### Author Rebuttal · Reviewer_WdW4 · 2026-04-02
> >
> > Thank the authors for their detailed responses to my concerns in the rebuttal. The authors have provided fairly thorough clarifications addressing each of the  main issues I raised.

---

> > > ### Author Response · Authors · 2026-04-03
> > >
> > > Dear Reviewer WdW4,
> > >
> > > Thank you for your invaluable time and effort in reviewing our paper. We are delighted to note that your concerns have been addressed in the rebuttal. Could you improve your scores to support our work? Thanks again.
> > >
> > > Sincerely yours,
> > >
> > > The Authors

---

### Official Review · Reviewer_KA8w · 2026-03-13

**Soundness:** 3
**Presentation:** 3
**Significance:** 3
**Originality:** 3
**Overall Recommendation:** 3
**Confidence:** 3

**Summary:**

The paper introduces Content-aware Speculative Decoding (CSD), a framework designed to accelerate the generation of images by leveraging the principles of Speculative Decoding (SD). Unlike previous methods that apply acceleration uniformly across an image, CSD recognizes that different regions of an image contain varying levels of complexity.

**Compliance With Llm Reviewing Policy:**

Affirmed.

**Final Justification:**

Most of my concerns are solved. However, I discover some new issues in the author's reply to WdW4:

First, MaskGIT (NAR) can be the generation order of AR, and the corresponding AR position token can be retained each time.

In addition, NAR can be used for AR acceleration design [1].

Therefore, I modified my score.

[1] Resurrect mask autoregressive modeling for efficient and scalable image generation.

**Key Questions For Authors:**

1. Computational Overhead: What is the specific wall-clock latency cost for calculating entropy and the TV-distance filter at each step? Does this overhead noticeably offset the theoretical gains from speculative decoding?

2. High-Complexity Scenes: How does CSD perform on images lacking “smooth regions” (e.g., dense textures or high-entropy patterns)? Does the acceleration advantage diminish or revert to standard SD levels in these cases?

**Limitations:**

Yes.

**Strengths And Weaknesses:**

Strengths:

1. Optimal Resampling: The authors don’t just relax the acceptance criteria; they provide a proof that the resampling distribution used in their framework remains optimal under probabilistic relaxation, ensuring that the statistical integrity of the target model is preserved.

2. Clear Framework Structure: The division of the methodology into two distinct modules (Entropy-based relaxation and TV-distance filtering) makes the technical approach easy to follow.

Weaknesses:

1. Entropy Calculation Overhead: While the paper mentions efficiency, the “cost” of calculating the entropy of the target model at each step is not explicitly detailed in the snippet. If the entropy calculation is computationally expensive, it might partially offset the speed gains of speculative decoding.

2. Minor Redundancy: In the introductory section (Lines 037–045), the text repeats the sentence: “The post-processed acceptance probabilities are highly correlated with target model entropy, achieving content-aware speculative decoding” almost verbatim twice.

---

> ### Author Rebuttal · Authors · 2026-03-31
>
> We sincerely thank the reviewers for recognizing the theoretical contribution of our optimality proof and the clarity of our framework design. Below we response the concerns point-to-point:
>
> ### W1. Entropy Calculation Overhead
> Thank you for raising this important concern about the computational overhead of entropy calculation. To directly address it, we measured the average per-iteration decoding latency on Janus-Pro-7B using an MS-COCO 1000-subset. The results confirm that **the entropy computation overhead is negligible**.
>
> Specifically, the average per-iteration latency of our CSD method is **35.46 ms**, while the entropy calculation itself takes **only 0.038 ms**. This accounts for **merely ~0.1%** of the total iteration time, indicating that the overhead is practically insignificant.Therefore, the entropy-guided relaxation in CSD does not compromise the decoding efficiency. Instead, it preserves the speed gains of speculative decoding with only marginal additional cost.
>
> | Component | Latency (ms) | Proportion of Per-iteration Latency (%) |
> |--------|---------------------------|----------------------|
> | Per-iteration Latency (total) | 35.46 | 100 |
> | Entropy Calculation | 0.038 | 0.11 |
> | TVD Calculation & Filtering | 0.020 | 0.06 |
>
> ### W2. Minor Redundancy
> We appreciate the reviewer's attention to detail. The duplicate sentence will be removed in the revised version to eliminate redundancy.
>
> ### Q1. Computational Overhead
>
> We thank the reviewer for the important question regarding wall-clock overhead. To directly evaluate this overhead, we also measured the average per-iteration decoding latency on Janus-Pro-7B using an MS-COCO 1000-subset. The results show that **the overhead of computing entropy and the TV-distance filter is negligible and does not offset the gains of speculative decoding**.
>
> On the one hand, CSD adds only **0.06ms** per iteration—a mere **0.17%** computation increase, compared to SJD. This marginal cost comes from computing the entropy of the target model's output distribution, which involves a simple dot product and log operation over the vocabulary dimension.
>
> On the other hand, CSD achieves lower computation overhead than GSD **(35.46ms vs. 37.24ms)**, as GSD relies on clustering-based grouping that requires sorting and distance computations. This highlights that entropy-based content adaptation is **lightweight computing ** compared to alternative content-aware strategies.
>
> | Method | Per-iteration Latency (ms) | Overhead vs SJD (ms) | Overhead vs SJD (%) |
> |--------|---------------------------|----------------------|----------------------|
> | SJD (baseline) | 35.40 | — | — |
> | GSD | 37.24 | +1.84 | +5.20% |
> | CSD (ours) | 35.46 | +0.06 | +0.17% |
>
> ### Q2. High-Complexity Scenes
>
> Thanks for this insight comment. In high-complexity scenes without smooth regions, CSD will **dynamically revert to standard speculative decoding**, ensuring no quality degradation. To empirically validate this adaptive behavior, we tested on a high-complexity prompt where smooth regions are minimal:
>
> Prompt:
> > High-tech biological fusion texture, surface of a supercomputer covered in moss and fungi, but the moss grows in perfectly intricate circuit trace patterns, organic decay meets digital precision, dense clustering, hyper-detailed, macro photography, moody lighting with neon green and deep purple, complex layering, 8k.
>
> We measure the average target model entropy across generated tokens: 3.1, which is significantly lower than the smooth image example shown in Figure 1. We found that CSD's distribution alignment filter frequently exceeds the threshold $\delta$, causing $\epsilon = 0$ for most positions. Consequently, the acceptance probability reverts to $min(1, q/p)$ , matching standard speculative decoding. The above generated images and discussion will be added in our revision, as this rebuttal cannot provide the image materials.

---

> > ### Author Rebuttal · Reviewer_KA8w · 2026-04-02
> >
> > Most of my concerns are solved. However, I discover some new issues in the author's reply to WdW4:
> >
> > First, MaskGIT (NAR) can be the generation order of AR, and the corresponding AR position token can be retained each time.
> >
> > In addition, NAR can be used for AR acceleration design [1].
> >
> > Therefore, I modified my score.
> >
> > [1] Resurrect mask autoregressive modeling for efficient and scalable image generation.

---

> > > ### Author Response · Authors · 2026-04-02
> > >
> > > Thank you for your feedback and for engaging in this technical discussion.
> > >
> > > Regarding your point: Several Non-Autoregressive (NAR) models, such as MaskGIT, can be configured to generate tokens in an autoregressive (AR) order (i.e., token by token) while preserving positional information. these approaches essentially abandon their inherent parallel generation capability and fundamentally alter their underlying decoding strategy. Consequently, for the sake of clear classification and to maintain a distinct differentiation from pure AR models, we still categorize these models as NAR, which is consistent with the original definition provided in MaskGIT itself [2].
> > >
> > > Concerning work [1] leverages a NAR model to accelerate an AR pipeline by first generating a small set of tokens with AR and then filling the remaining tokens via a separate NAR model. In contrast, our CSD method accelerates mainstream AR models (e.g., Lumina-mGPT and Janus-Pro) directly without the need for any additional models or auxiliary pipelines. Given this fundamental difference in mechanism, our claim that "parallel generation capability cannot be directly transferred to autoregressive generation without fundamentally altering the model architecture" remains correct.
> > >
> > > Our method represents an innovation in speculative decoding, including entropy-based probability relaxation and distribution alignment filter. As such, neither MaskGIT nor [1] affects the primary contributions and novelty of our approach.
> > > If any inaccuracies remain, we welcome further discussion.
> > >
> > > [2] MaskGIT: Masked Generative Image Transformer. CVPR 2022.

---

### Official Review · Reviewer_9B7J · 2026-03-13

**Soundness:** 3
**Presentation:** 3
**Significance:** 3
**Originality:** 3
**Overall Recommendation:** 4
**Confidence:** 3

**Summary:**

The paper proposes CSD，a framework designed to accelerate autoregressive image generation. It introduces an entropy-based probability relaxation mechanism that adjusts acceptance criteria based on regional information density. By relaxing constraints in low-uncertainty areas and employing a TV-distance-based filter, the method aims to increase the speculative token acceptance rate while maintaining image quality across models like Janus-Pro and Lumina-mGPT.

**Compliance With Llm Reviewing Policy:**

Affirmed.

**Key Questions For Authors:**

See Limitations

**Limitations:**

yes

**Strengths And Weaknesses:**

####  Strengths
* **Novel Perspective:** The paper approaches the problem from a very good angle by considering content-awareness in speculative decoding, specifically focusing on how different image regions (background vs. detail) impact the generation process.
* **High Reproducibility:** The source code is provided and open-sourced, which ensures high reproducibility and is beneficial for the research community.

####  Limitations
* **Lack of Novelty:** Using entropy as a confidence measure for estimation is a very common technique. It has been extensively used in NLP and is also widely applied in image quality assessment, making the core methodological contribution appear incremental.
* **Marginal Performance Gains:** The actual effectiveness of the proposed method is not particularly strong. Comparative experiments indicate that the performance improvement over existing baselines is limited.
* **Over-reliance on Heuristic Thresholds:** The performance of the content-aware relaxation heavily depends on manually tuned hyperparameters (such as the entropy threshold). The paper lacks a robust analysis of how these thresholds generalize across diverse datasets or different base model architectures, which might limit its practical utility in varying scenarios.

---

> ### Author Rebuttal · Authors · 2026-03-31
>
> We sincerely thank the reviewers for recognizing the novelty, and the reproducibility of our work. Below we response the concerns point-to-point:
>
> ### W1. Concerns on the Entropy
> We agree that entropy is a widely used uncertainty metric in NLP. However, it is just an uncertainty metric tool, which is used to explore an interesting finding to propose a novel AR model acceleration method for image generation. In details, our main contributions are threefold:
>
> - We observe **a strong correlation between target model entropy and image region characteristics**: smooth, low-texture regions exhibit high entropy, while high-frequency details show low entropy. This insight reveals that entropy reflects generation difficulty in AR image models, enabling us to accelerate simple regions more aggressively without harming visual quality.
>
> - We provide a **theoretical proof that the resampling distribution remains optimal under probabilistic relaxation**, and integrate entropy as a dynamic signal into the speculative decoding framework. This enables our method to achieve both optimality and content-awareness.
>
> - We complement entropy-based relaxation with **a distribution alignment filter based on TV distance**, which ensures that probabilistic relaxation is applied to effectively preserving generation fidelity, when draft and target distributions are well-aligned.
>
> ### W2. Marginal Performance Gains
> We argue, with respect that our CSD achieves **substantial rather than marginal gains**. In Table 2 of the original manuscript, we intentionally report the results at a consistent acceleration rate to show that CSD achieves better quality (CLIP 32.28 vs 32.16, FID 33.69 vs 34.00). This advantage is also clearly reflected in the Pareto-front shown in Figure 5.
>
> Furthermore, CSD’s advantage becomes more pronounced when we prioritize the speed. We also conducted additional experiments on Janus-Pro. By relaxing the TV threshold $\delta$ to 0.8, CSD achieves **4.33x latency acceleration (5.30s)** while maintaining competitive quality (CLIP 32.21, FID 33.58). As shown in the table below, our CSD reduces **24% and 26% inference time** than GSD (6.93s) and Addition (7.13s), with the best CLIP/FID scores across all baselines.
>
> | Method | Latency($\downarrow$) | Step($\downarrow$) | FID($\downarrow$) | CLIP score($\uparrow$) |
> | :--- | :--- | :--- | :--- | :--- |
> | Amplify($k=4$) | 7.05s (3.26x) | 165.7 (3.48x) | 34.01 | 32.20 |
> | Addition($\epsilon=0.4$) | 7.13s (3.22x) | 162.8 (3.54x) | 33.96 | 32.16 |
> | GSD($G=35$) | 6.93s (3.31x) | 161.7 (3.56x) | 34.00 | 32.16 |
> | Ours($\lambda=0.5, \delta=0.8$) | 5.30s (4.33x) | 134.3 (4.29x) | 33.58 | 32.21 |
>
> ### W3. Over-reliance on Heuristic Thresholds
>
> We thank the reviewer for raising this important concern. Our analysis reveals generalized patterns: the relaxation coefficient $\lambda$ is stable, while the optimal value of TV-distance threshold $\delta$ depends on model capacity and transfers consistently across datasets.
>
> **The effect of $\lambda$**: The optimal choice is stable and can be fixed without extensive tuning. As shown in Table 3, $\lambda$ = 0.5 consistently yields a good trade-off between acceleration and quality, while larger $\lambda$ leads to noticeable performance degradation and smaller one decreases the acceleration rate with the same CLIP score.
>
> **The effect of $\delta$**: This hyperparameter is of critical importance, as the model architecture exerts significant influence on both performance and inference speed. A larger $\delta$ yields high acceleration but at the expense of performance, whereas a smaller $\delta$ results in lower acceleration but superior performance stability.
>
> Consequently, we propose a search strategy starting with determining the target performance. For high performance, we aim for 0.35, and for high acceleration, 0.65. Then use a search with 0.1 interval based on results. Generally, tuning shouldn't exceed four times.
>
> Moreover, we also found that the optimal $\delta$ exhibits consistent behavior across different datasets. For instance, the $\delta$ values validated on MS-COCO transfer well to the GenEval benchmark.
>
> | Method | Latency($\downarrow$) | Step($\downarrow$) | FID($\downarrow$) | CLIP score($\uparrow$) |
> | :--- | :--- | :--- | :--- | :--- |
> | Vanilla AR (Janus-Pro 1B) | 15.29s (1.00x) | 576 (1.00x) | 32.75 | 32.05 |
> | SJD  | 7.86s (1.95x) | 303.8 (1.90x) | 32.49 | 32.05 |
> | Ours($\lambda=0.5, \delta=0.35$) | 7.10s (2.15x) | 270.9 (2.13x) | 32.63 | 32.04 |
> | Ours($\lambda=0.5, \delta=0.65$) | 4.56s (3.35x) | 173.0 (3.55x) | 32.41 | 32.00 |
>
> | Method | Latency($\downarrow$) | Step($\downarrow$) | Overall($\uparrow$) |
> | :--- | :--- | :--- | :--- |
> | Vanilla AR (Janus-Pro 7B) | 22.95s (1.00x) | 576 (1.00x) | 0.79 |
> | SJD | 11.95s (1.92x) | 293.6 (1.96x) | 0.79 |
> | Ours($\lambda=0.5, \delta=0.65$) | 6.84s (3.36x) | 161.7 (3.56x) | 0.79 |
> | Ours($\lambda=0.5, \delta=0.8$) | 5.53s (4.15x) | 130.8 (4.40x) | 0.78 |

---

> > ### Author Rebuttal · Reviewer_9B7J · 2026-04-04
> >
> > My concerns have been adequately addressed.

---

> > > ### Author Response · Authors · 2026-04-04
> > >
> > > Dear Reviewer 9B7J,
> > >
> > > Thank you again for your invaluable time and the effort on our paper. Thank you very much for approving our work!
> > >
> > > Sincerely Yours,
> > >
> > > The Authors

---

### Decision · Program_Chairs · 2026-04-30

**Decision:**

Accept (regular)

**Comment:**

This paper aims to accelerate autoregressive image generation through content-aware speculative decoding. The issues were largely resolved by the rebuttal, and the 3 of the 4 reviewers marked the rebuttal as fully resolved their concerns. Reviewers' consensus became mostly positive after the rebuttal. While there are still some concerns about novelty and positioning, the method is sound, effective, and relevant to the community, and I therefore support acceptance.